# Use of Biostimulants to Alleviate Anoxic Stress in Waterlogged Cabbage (*Brassica oleracea* var. *capitata*)—A Review

Nadya Buga [1],* and Marko Petek [2]

[1] University of Zagreb Faculty of Agriculture, 10000 Zagreb, Croatia
[2] Department of Plant Nutrition, University of Zagreb Faculty of Agriculture, 10000 Zagreb, Croatia; mpetek@agr.hr
* Correspondence: nadya.alexandra.buga@gmail.com

**Abstract:** In Europe, the intensity and frequency of flooding events are expected to increase due to climate change, adding additional challenges to agricultural production and creating the need for new plant products and adaptation tools. Cabbage is one economically important vegetable that is likely to be affected by increased flooding in Europe. This review investigates the potential of biostimulant applications based on algae extracts, amino acids, microorganisms, and nano-$CaCO_3$ to assist cabbage plants subjected to waterlogged conditions. The overall findings from the studies reviewed indicate that these biostimulants could aid plants suffering from anoxic stress due to waterlogging through their ability to improve nutrient availability and plant nutrient status, modulate phytohormones and phytohormone signalling, increase compatible solutes, and enhance the antioxidant system. The effect of biostimulants is influenced by multiple factors; therefore, field studies are required to determine the most valuable biostimulant combination and application dose, type, and timing for cabbage, as well as any economic benefits that could arise. More precise information would benefit food producers by providing them with additional adaptation tools to use in a changing climate as well as natural products that are compatible with the agriculture sector's transition to more sustainable and ecological management.

**Keywords:** flood; nutrients; oxidative stress; sustainable agriculture; vegetable production





## 1. Introduction

Alongside warming and changing precipitation patterns, climate change is marked by an increase in the intensity and frequency of extreme events such as heatwaves, droughts, and heavy precipitation [1]. Heavy rainfall events, and their increasing intensity and frequency due to climate change, constitute a serious threat to agriculture, as soil flooding negatively impacts plant growth and can lead to plant death [2]. There is an increasing amount of observational evidence that confirms an increase in heavy rainfall events in Europe of approximately 45% in the years 1981–2013 compared to 1951–1989 [3]. This information gives reason to investigate measures that could aid in the protection of crops, particularly at vulnerable stages of development, against heavy precipitation events in the European region.

*Brassica* crops are among the 10 most economically important vegetable crops globally [4]. Cabbage is one of the most popular *Brassica* vegetables and has many beneficial properties for human health, such as reducing the risks of cancer, cardiovascular disease, Alzheimer's disease, and diabetes [5], hence the desire to protect cabbage production. In 2021, Europe produced 9.3 million tonnes of cabbage across 300,000 hectares, with the largest producers being the Russian Federation, Ukraine, Germany, Poland, and Romania [6].

Cabbage grows well in a cool, humid climate and has a growing period of 90–200 days, depending on variety and growing conditions [7]. Water requirements vary from 380–500 mm; however, young cabbage seedlings are very sensitive to excess soil moisture, and cabbage

generally requires well-drained soil for optimal growth [8,9]. Casierra-Posada and Cutler [10] found cabbage to show little tolerance to 25-day flooding, which caused significant changes in growth parameters and chlorophyll content. A more recent study concluded that small cabbage plants are not highly sensitive to short-duration waterlogging (e.g., 3 days long), only showing signs of stress after repeated waterlogged conditions [11]. As the duration of waterlogged conditions varies year to year and is expected to increase, strategies for dealing with these conditions are necessary to protect cabbage production when crops become stressed.

There is a growing body of research that supports the use of biostimulants to enhance plant growth and development, nutrient efficiency, stress tolerance, and crop quality traits [12–14]. Biostimulants are agrochemical products that are formulated from natural substances and/or microorganisms and improve plant health/development in some way [13]. They can be applied as a foliar spray, a soil drench, or both; the choice of application will depend on a variety of factors, including environmental conditions, time of application, and plant species/cultivar [15]. Biostimulants differ from fertilizers in that they are only applied in small quantities and do not contain sufficient quantities of nutrients by themselves, but instead enhance nutrient uptake by the root system and transport within the plant [12,14]. This function is especially important during flooding, when nutrient uptake might become inhibited and eventually result in reduced yields [16]. Additionally, biostimulants have the potential to offer a natural alternative to synthetic products, thereby reducing environmental pollution, and their use in agriculture is predicted to become standard practice in the coming years as this sector moves towards more ecological management [14].

Biostimulant application to crops has been reported to stimulate vegetative growth, improve nutrient uptake efficiency and distribution, increase tolerance to biotic and abiotic stress, improve plant vigour, increase antioxidant capacity, and ultimately enhance crop yield and quality [13]. The 5th Biostimulant World Congress reported nine categories of substances that act as biostimulants: (1) Humic substances, (2) complex organic materials, (3) beneficial chemical elements, (4) inorganic salts, (5) algae extracts, (6) derivatives of chitin and chitosan, (7) antiperspirants, (8) free amino acids and N-containing substances, and (9) plant growth-promoting rhizobacteria (PGPR), arbuscular mycorrhizal fungi (AMF), and *Trichoderma* spp. [13]. As the biostimulant landscape continues to evolve, various other products, such as natural extracts, are being discovered and utilized as biostimulants. For example, natural extracts from fennel and ammi seeds, moringa leaf, or honey have been found to produce beneficial effects against abiotic stress when applied to crops by stimulating the antioxidant system [17–19].

This review focuses on more conventional biostimulants based on algae extracts, amino acids, and beneficial microorganisms, as well as delving into the potential of nanoparticle biostimulants such as nano-calcium carbonate. A wide range of literature investigates the effects of biostimulant application on various crops experiencing biotic or abiotic stress. However, when considering abiotic stress conditions, more studies tend to focus on drought, salinity, or heavy metal exposure, and there is less information available on the effect of biostimulants on flooded or waterlogged crops. This review aims to fill this knowledge gap by investigating specifically whether biostimulants show promise against waterlogging stress. Findings to support biostimulant use under these conditions could provide producers with a useful adaptation tool to protect plants against the negative effects of climate change. Additionally, if biostimulants are useful against abiotic stress from waterlogging, these natural and ecological products could aid producers in the transition to a more sustainable production system. A major concern with waterlogging is oxidative stress and the impairment of the plant's antioxidant system that results from anoxic conditions; therefore, this review explores the relationship between flooding, oxidative stress, and biostimulant use.

## 2. Flooding and the Effect on Nutrients

Overwet environments create stress for plants by significantly slowing gaseous diffusion and preventing aerobic mechanisms that are vital for plant health [16]. 'Flooding' is a general term for excessively wet conditions, and care must be taken to use consistent terminology for the type of excessive water conditions studied in an experiment. Four types of flooding have been suggested for use in flooding and low $O_2$ stress research [20]: (1) "waterlogging", in which only the root-zone is flooded; (2) "partial waterlogging", in which the root-zone is partially flooded; (3) "submergence", in which the entire plant (root and shoot) is under water; and (4) "partial submergence", in which the entire root system and only part of the above-ground organs are under water.

Plants generally respond in the following ways to flooding: Overall decrease in shoot growth, inhibition of leaf growth, inhibition of stem extension, and inhibition of photosynthesis (slower increase in dry weight) [16]. Under ordinary soil conditions, root growth and function are facilitated by aerobic respiration, in which $O_2$ is supplied from the rooting environment and acts as the terminal electron acceptor in the electron transport chain (ETC) [16]. Excessive rainfall can create conditions that lead to root hypoxia or anoxia and cause plants to switch to anaerobic metabolism, which negatively impacts root growth and activity [2,21,22]. Anaerobic metabolism is unable to provide sufficient energy for ion pumps and causes nutrient uptake via roots and radial transport to become immediately inhibited in non-wetland species, which subsequently reduces the quality and yield of crops [16,23]. The severity of flooding impact depends on the duration of the flood, the type of flooding, the climatic conditions, the growth stage of the plant, and the plant species/cultivar [16]. When only the root system is flooded (i.e., waterlogging), flooding effects on the shoot result from changes in the internal flow of substances (water, photosynthate, inorganic nutrients, hormones or precursors, and toxins) between the root and shoot [16]. The soil environment also changes during waterlogging: some nutrients become less available (N and S), some become more available (P, Fe, Zn, and Mb), and there is an increase in the accumulation of ammonium and polyphenolic compounds and the rate of denitrification [21].

The extent of the waterlogging effect on cabbage is influenced by the duration of flooding and the developmental stage of the plant, as can be seen from the following experiments with cabbage (*Brassica oleracea* var. *capitata*). At an early growth stage, 24 h flooding reduced growth temporarily, but there was no difference in yield with the control during harvest [24]. Flooding of 6 h, 12 h, and 24 h duration at a later growth stage led to a decreased yield (averaging 3.1%, 6.4%, and 9.5%, respectively, across a 2-year experiment) [24]. Cabbages at the stage of 3–5 leaves exposed to short (3-day) or repeated waterlogged conditions showed decreased proline and ABA and increased salicylic acid (SA) [11]. However, once waterlogged conditions were removed, these parameters returned to normal. Regarding nutrients, P, K, Mg, Zn, and Cu significantly decreased due to waterlogging (varying for 3-day or repeated waterlogging), while Mn and Ca significantly increased [11]. Cabbages at the stage of 2–3 leaves exposed to 25-day waterlogging showed significant reductions in the leaf area, total dry weight, chlorophyll content, chlorophyll fluorescence, leaf area ratio, and growth rate, which the authors took to indicate a low waterlogging tolerance [10]. In the 3-day and 25-day waterlogging experiments, the cabbages were not grown to completion and harvested, so it is not known how these early symptoms of stress might correlate to changes in harvestable yield. Furthermore, it can be seen from the studies above that experiments investigating waterlogging effects on cabbage have not used consistent methodology, so it is difficult to conclude exact thresholds for waterlogging sensitivity and tolerance.

### *Brassica oleracea and Oxidative Stress*

Similarly to flooding, several -oxic terms have been suggested for distinguishing different oxygen conditions during research: (1) "normoxia", the reference condition for normal $O_2$ availability in air; (2) "hypoxia", in which $O_2$ concentrations are below normoxic and specific processes might be affected; (3) "anoxia", which describes a complete absence

of $O_2$ in a system; and (4) "hyperoxia", in which $O_2$ concentrations are above normoxia [20]. Periods of heavy precipitation can lead to anoxic stress for many intolerant species, such as *Brassica oleracea* [25]. Waterlogging-induced hypoxic or anoxic conditions impair plant metabolism and can result in oxidative stress, which occurs through the overaccumulation of reactive oxygen species (ROS) free radicals (superoxide anion, hydroperoxyl radical, alkoxy radical, and hydroxyl radical) and nonradical molecules (hydrogen peroxide and singlet oxygen) [22,26,27]. ROS interact with phytohormones and play an important role in stress signalling pathways; however, the build-up of ROS causes damage to important cellular components such as carbohydrates, proteins, lipids, and DNA [28,29].

Excessive ROS accumulation is considered one of the most crucial consequences of abiotic stress, resulting from disequilibrium between the generation of ROS and their detoxification by the antioxidant defence system, which comprises enzymatic and nonenzymatic antioxidants [28]. Enzymatic antioxidants include superoxide dismutase (SOD), catalase (CAT), ascorbate peroxidase (APX), glutathione reductase (GR), monodehydroascorbate reductase (MDHAR), dehydroascorbate reductase (DHAR), glutathione peroxidase (GPX), guaiacol peroxidase (GOPX), glutathione S-transferase (GST), ferritin, nicotinamide adenine dinucleotide phosphate (NADPH), oxidase-like alternative oxidase (AOX), peroxiredoxins (PRXs), thioredoxins (TRXs), and glutaredoxin (GRX) [28]. Nonenzymatic antioxidants include ascorbic acid (AsA), glutathione (GSH), phenolic acids, alkaloids, flavonoids, carotenoids, $\alpha$-tocopherol, and nonprotein amino acids. These antioxidants work in a coordinated manner to directly or indirectly scavenge ROS and/or inhibit their overproduction in the ascorbate-glutathione (AsA-GSH) cycle [28].

Several crop species can survive waterlogging conditions for various durations by activating their antioxidant defence systems [28]. For example, in maize, higher activity of SOD, peroxidase (POX), and CAT was noted under waterlogging conditions [30]. When oxygen is unavailable and unable to act as the terminal electron acceptor in the electron transport chain (ETC), intermediate electron carriers become reduced, which interferes with redox-active metabolic reactions [22]; the antioxidant system can become impaired, with limited recycling, de novo synthesis, and transport of antioxidants. De novo synthesis of antioxidants relies on a sufficient energy supply, which cannot be sustained during anoxic conditions [22]. This is where the antioxidant potential of biostimulants becomes useful.

## 3. Algae Extract Biostimulants

Some biostimulants, e.g., ExelGrow, are based on algae extracts. *Ascophyllum nodosum* is a brown inter-tidal seaweed that is the most widely researched seaweed in terms of biostimulant use [31]. *A. nodosum* extracts (ANEs) are a source of phytohormones (abscisic acid, cytokinins, auxins), organic osmolites, macro- and micronutrients, vitamins, amino acids, and several bioactive compounds: Poly- and oligosaccharides (laminaran, fucan, alginate), antioxidants, peptides, betaines, and secondary metabolites (sterols) [32,33]. The precise composition of ANEs depends on the extraction method used, e.g., water-based extraction, acid or alkaline hydrolysis, microwave-/ultrasound-/enzyme-assisted extraction, super-critical fluid extraction, and pressurised liquid extraction [31]. Algae biostimulants can be applied as a foliar spray, soil application, or both [34]. Some studies have reported foliar application to be more effective since foliar absorption occurs almost instantly and particle mobility is not inhibited by adsorption to soil particles [34]. Although other studies report equal effectiveness for both types of applications [35].

ANEs are able to regulate molecular, physiological, and biochemical processes in plants, improve plant growth and defence, and mitigate some abiotic and biotic stresses [31]. Generally, the effects of ANEs on plants have been attributed to their content of phytohormones, micronutrients, alga-specific polysaccharides, betaines, polyamines, and phenolic compounds [32]. In a study evaluating the effect of ANE application during various stress conditions (drought, salinity, freezing, and biotic stress), ANEs were found to maintain photosynthesis, modulate hormonal homeostasis, and alter nutrient uptake and assimilation by regulating the transcription of transporters [32].

Several modes of action are proposed for ANEs (Figure 1); for *Brassica oleracea*, the mode of action relates to isothiocyanate, phenolic compounds, and flavonoid compounds. Lola-Luz et al. [33] found that the application of *A. nodosum* extract to cabbage grown under normoxic conditions increased the content of phenolic and flavonoid compounds in *B. oleracea*. In this experiment, two ANEs, AlgaeGreen™ and XT, were applied as a foliar spray to cabbages once per month at application rates of 3.5 L ha$^{-1}$ (containing 2.6 mL seaweed extract) and 5 L ha$^{-1}$ (containing 3.7 mL seaweed extract) until harvest. Cabbages were harvested 15 days after the final application. However, this increase in phenolic and flavonoid compounds did not translate to a significant increase in total yield compared to the untreated control, despite this effect being reported in other studies. Perhaps the beneficial effects of biostimulants are more noticeable in crops under stress.

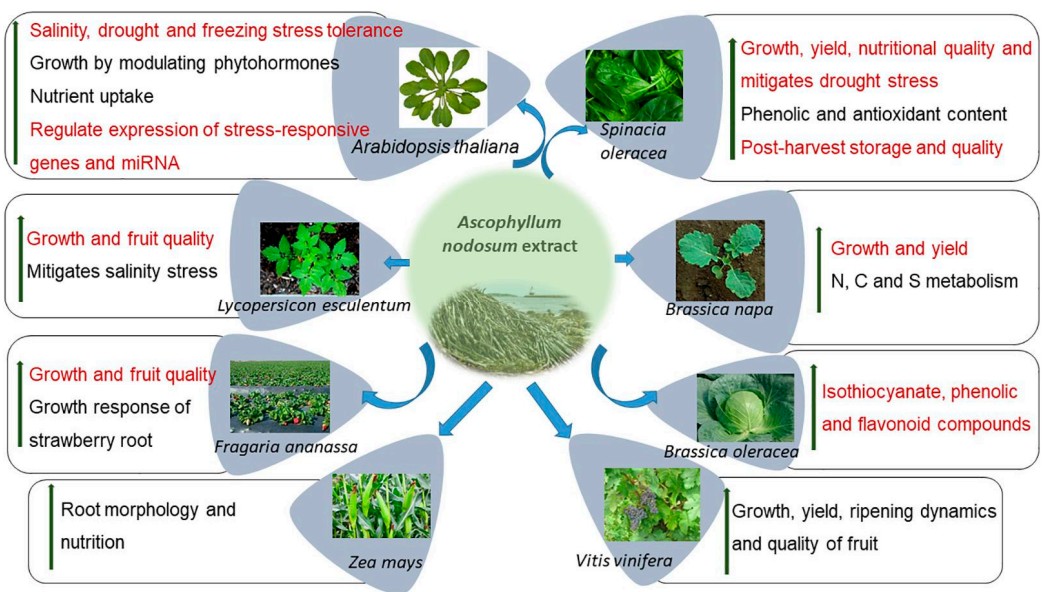

**Figure 1.** *Ascophyllum nodosum* extract (ANE) improves the growth of several crops by different modes of action [31].

### 3.1. Effect of Ascophyllum nodosum on Nutrient Acquisition

ANE applications have been found to increase nutrient availability and uptake, and nutrients from ANEs can be readily absorbed by leaves through stomata and cuticle hydrophilic pores [31,36]. This foliar absorption would be particularly useful when plants are waterlogged and nutrient acquisition through the roots is inhibited. Several studies report an increase in nutrients following ANE application, although the effect varies for different crops. In salinity-stressed tomato, *A. nodosum* increased the content of N, P, K, Ca, S, Mg, Zn, Mn, and Fe [37]; in olive, only K, Fe, and Cu increased [38]; in broccoli, there were increases in dry matter, yield, and content of P, K, Ca, and Mg [39]. Evidence suggests that crops have a very individual response to biostimulant applications, and the biostimulant effect will depend on crop species as well as many other factors. These nutrient increases could be beneficial for cabbage and other crops experiencing longer waterlogged conditions to allow for some nutrient acquisition to still occur, even while nutrient uptake by the roots is completely inhibited.

Mancuso et al. [40] believe entry through the stomata to be the preferential route for K$^+$ and Ca$^{2+}$ ion uptake and additionally suggest that chelating compounds in seaweed (e.g., mannitol) are responsible for increasing nutrient availability and absorption. Other studies suggest that the increased nutrient availability and uptake are likely due to changes in the regulation of genes involved in nutrient acquisition [31]. Through their microarray analysis, Jannin et al. [41] identified approximately 1000 genes whose expression was significantly affected by the application of seaweed extract to *Brassica napus*. The most affected pathways involved carbon and photosynthesis, cell metabolism, and N and S responses to stress.

Specifically, they found the enhancement of genes that encode proteins involved in the uptake and assimilation of N and S.

Some studies have noted that the beneficial effects of ANE application are dose dependent, with the best results being obtained on plants receiving a medium dosage [36,42]. In their experiment, da Silva et al. [42] tested 0.0, 2.0, 4.0, and 6.0 mL/L doses of seaweed extract on young collard greens (*Brassica oleracea* L.) grown in substrate of steep bank (75%), and cattle manure (25%), with 3 applications carried out at 6-day intervals. Analysis was conducted 23 days after sowing; based on graphical interpolation, the authors concluded that a 3.80 mL/L application of *A. nodosum* extract would result in the most improvements to early development growth parameters as compared to no application or a higher application. Similarly, Hidangmayum and Sharma [36] found that onion growth parameters increased the most with a 0.55% concentration treatment of pure liquid seaweed, while higher concentrations showed a decreasing trend. Extensive experiments will need to be conducted in order to determine the optimum dose of different biostimulant products for different crop species under various conditions.

*3.2. Protection from Abiotic Stress (Waterlogging)*

One type of algae biostimulant protection from abiotic stresses stems from increased antioxidant activity [12,14]. Plant tissues contain several enzymes and products that control the level of ROS and protect cells from stress; these include scavenging ROS enzymes, detoxifying lipid peroxidation (LP) products, and low-molecular-mass antioxidants (ascorbate, glutathione, phenolic compounds, and tocopherols) [22]. ANE application has been found to increase glutathione reductase activity and enhance levels of ascorbate and glutathione in stressed plants [43,44]. Glutathione has a reduced form (GSH) and an oxidised form, glutathione disulfide (GSSG); they act together to maintain redox balance in cellular compartments [22]. Furthermore, through the AsA-GSH cycle, GSH can regenerate the water-soluble antioxidant ascorbic acid. Ascorbic acid is the main compound in the aqueous phase to detoxify ROS, as it is able to donate electrons in a variety of enzymatic and nonenzymatic reactions [22]. This suggests that when applied foliarly to waterlogged cabbage plants, algae biostimulants may have the ability to help protect plants from damage caused by oxidative stress.

Jannin et al. [41] report on multiple studies of seaweed extract application to crops such as grapevine, strawberry, soybean, tomato, and maize. These studies found that seaweed extract application accelerated crop development cycles, increased total dry weight, increased proliferation of secondary root systems, enhanced leaf chlorophyll content, improved development, and increased crop growth (leading to increased yield, quality, and size of harvested products). In the studies, the authors suggest that phytohormones, such as auxins or cytokinins contained in the seaweed extract, are likely responsible for these effects. Jannin et al. [41] propose that the beneficial effects may be the result of several components (phytohormones, betaines, polymers, and nutrients) working together synergistically. The enhancement of these factors, which are inhibited under waterlogged conditions, indicates that algae biostimulants could, in theory, mitigate some of the negative effects experienced by waterlogged crops.

## 4. Microorganism and Amino Acid Biostimulants

Some biostimulants, e.g., Organico, are based on beneficial microorganisms and amino acids. Products containing these substances are said to have stimulating, bacterial, and fungicidal effects, activate the plant's natural defence system, protect plants from extreme weather and disease, accelerate development, and increase the speed of reaction to stressful conditions and disease [45]. *Bacillus* species, in particular, are popular bacteria used as biostimulants [46].

### 4.1. The Effect of Bacillus spp. as Plant Growth Promoting Bacteria (PGPB)

Several *Bacillus* spp. have been found to suppress pathogens and promote plant growth, and for this reason, they have been considered plant growth-promoting bacteria (PGPB) [47]. PGPB are also known to change or release hormones in plants, produce plant growth-promoting volatile organic compounds, improve nutrient availability and uptake, and enhance abiotic stress tolerance [48]. Some mechanisms of action are shown in Figure 2. With regards to water stress, attention is drawn to phytohormone level modulation, increased antioxidant activity, osmolyte production, and the secretion of exopolysaccharides (EPS). While the majority of PGPB inhabit the rhizosphere and are isolated from the soil environment, some bacteria migrate aboveground and can be found on the aerial parts of plants, although in decreasing bacterial density compared to rhizosphere populations [49].

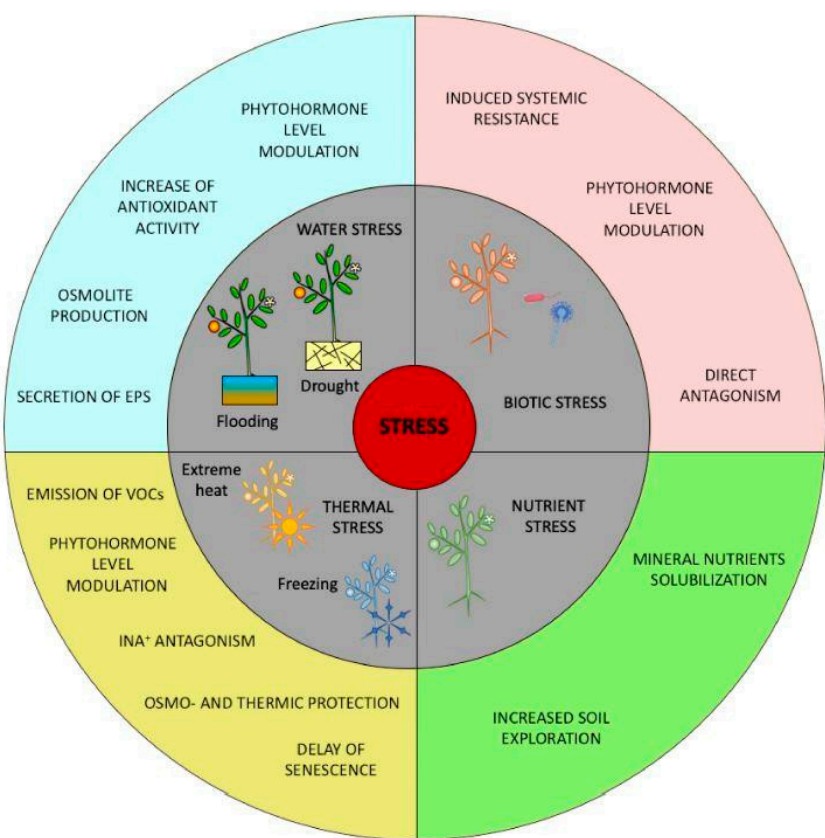

**Figure 2.** Schematic view of the protective mechanisms exerted by microbial biostimulants in relation to the stresses to which plants are subjected [50].

One method of using PGPB as a biostimulant in agricultural crops is through foliar application, in which a formula containing PGPB is sprayed onto the leaves of the plant [51]. Foliar applications are an important option if applying biostimulants during waterlogged conditions, when soil applications would become diluted. Foliar applications of PGPB (*B. subtilis*) have been found to stimulate plant growth, increase yield in apple varieties [52], and increase the photosynthetic rate of kale by up to 89% [53]. Efthimiadou et al. [51] also found foliar application of PGPB (*B. subtilis* and *B. megaterium*) to have a positive effect on the photosynthetic rate, yield, and dry weight of maize, but noted that soil application gave better results than foliar application. Further investigation could be carried out to understand which type of application is more suitable for waterlogged cabbage plants; whether foliar applications should be made during the flood event, or whether a foliar application or soil drench would be more effective post-flood.

*4.2. Effect of PGPB on Nutrients, Phytohormones and Oxidative Stress*

With regards to phyllosphere bacteria and foliar application, nitrogenase activity and indol-3-acetic acid (IAA) production have been considered the most important plant growth-promoting traits [54]. Phyllobacteria also stimulate plant growth through nitrogen fixation, phosphorous solubilisation, and siderophore production [54]. The foliar application of *B. subtilis* has been found to increase the activity of the ROS-scavenging enzymes polyphenol oxidase (PPO), SOD, and POX, which play an important role in alleviating oxidative stress in plants [55]. Another study found that the foliar application of *B. megaterium* under abiotic stress (drought) resulted in high expression of stress-related genes and increased content of total sugars, proteins, proline, phenolics, K, Ca, abscisic acid (ABA), and indole-3-acetic acid (IAA) [56]. It would be interesting to investigate whether similar effects are found under abiotic stress from excessively wet conditions; stimulation of ROS scavenging enzymes and increased phytohormone content would have positive effects against anoxic stress. Due to their ability to meet plant nutrient requirements, PGPB in the phyllosphere are being investigated for use as biofertilizers to promote crop growth [54]. In particular, the *Bacillus* genus was found to have a greater number of strains with plant growth-promoting traits. However, when considering microbial biostimulants, they are applied in too small quantities to be considered biofertilizers, although this does not negate the possibility of biostimulant applications having a positive effect on nutrient status.

Studies have investigated the effect of PGPB inoculations on *Brassica* spp. and found increases in nutrient status and phytohormone changes [57,58]. Inoculation of Chinese cabbage with *Bacillus subtilis* isolate resulted in significantly enhanced contents of K (6.48%) and S (45.9%) compared to the control and increased Mg (6.36%), Ca (11.5%), and P (73.3%) in the cell-free (CF) isolate treatment [58]. The JW1 strain of *B. subtilis* used in the experiment was also found to secrete higher amounts of exogenous gibberellins. PGPB inoculations have been found to improve the growth and development of host plants under stress conditions through the release of bioactive secondary metabolites including acetic acid, cytokinins, jasmonic acid, gibberellins, salicylic acid, and abscisic acid [58,59].

PGPB (*B. subtilis*, *B. megaterium*, and *Pantoea agglomerans*) applied as a soil drench to cauliflower resulted in increased gibberellic acid (23.6%), salicylic acid (89.5%), and indol-3-acetic acid (25.6%) [57]. Treated cauliflower plants also generally showed statistically significant increases in concentrations of N (2.9–22.9%), P (2.3–8.4%), K (2.7–12.0%), Mg (3.5–18.1%), Ca (2.2–8.5%), Zn (2.3–10.0%), Cu (−20.6–14.8%), Mn (0–20.1%), and Fe (6.9–31.7%). The range of nutrient concentrations reflects the variable effects resulting from different species and different strains. The highest content of P and Zn was obtained from *B. megaterium* TV-91C application, the highest Na, Cu, and Mn from *P. agglomerans* RK-92 application, and the highest N, K, Mg, and Fe from *B. megaterium* TV-87A application. The result for Cu stands out as only one strain of *B. megaterium* resulted in a lower value (−20.6%) compared to the control treatment; the other treatments improved Cu concentration. Aside from this low result, Na was the only other nutrient to be found in lower concentrations in four of the treatments. These results draw attention to the variable effects that can be expected from not only different species but also different strains, highlighting the fact that very detailed studies must be undertaken before conclusions can be drawn about the most beneficial treatment options.

The phytohormones mentioned above are plant growth regulators that are involved endogenously in various processes and stress responses. Auxins (e.g., acetic acid) play an important role in the development of adventitious roots and promote ethylene (ET) biosynthesis as a response to waterlogging, while ethylene production can promote the transport of auxin [59]. Cytokinins are involved in delaying stress-induced leaf senescence and inducing the accumulation of proline [60], which plays an important role during stress as a metal chelator, antioxidant defence molecule, and signalling molecule [61]. Jasmonic acid (JA) is involved in the defence response to abiotic stress [62]. Gibberellins (GAs) significantly increase under waterlogging in waterlogging-tolerant and waterlogging-resistant lines of soybean compared to waterlogging-sensitive lines [63] and are essential

for regulating the growth and development of plants [59]. Under flooding conditions, the accumulation of abscisic acid (ABA) in above-ground parts of plants improves their resistance to environmental stress [59].

Consistent with the beneficial effect of endogenous phytohormones, the foliar application of phytohormones has also been found to assist plant responses to abiotic stress. The exogenous application of JA can reduce damage to plants under waterlogging conditions and increase the content of ET, which aids in relieving waterlogging stress [59,62,64]. Spraying exogenous salicylic acid (SA) on peach trees was found to significantly increase the activities of ethanol dehydrogenase, POX, CAT, and proline in leaves and roots, which alleviated stress from waterlogging [65]. ABA application to *Arabidopsis thaliana* was found to initiate the antioxidant defence system, resulting in reduced oxidative damage and improved waterlogging tolerance [66]. These findings support the use of PGPB application to relieve waterlogging stress due to their ability to enhance phytohormones in stressed plants.

Another way plants respond to waterlogging is through the rapid accumulation of ethylene (ET) [59]. ET is produced by many plants as a generalised response to stress, and its production is reliant on the action of 1-aminocyclopropane-1-carboxylic acid (ACC) synthase and ACC oxidase [67]. Various stressors, including flooding, induce the transcription of ACC synthase genes. However, during anoxia, ACC oxidase is unable to complete the conversion of ACC (the precursor of ethylene) into ethylene since it is oxygen-dependent [50]. Instead, ACC is translocated from waterlogged roots to aerial parts of the plant, where it is converted to ethylene and causes wilting, leaf chlorosis/necrosis, flower/fruit drop, growth inhibition, reduced yield, or even death [68,69]. PGPB act as a biological sink for ACC since they possess the enzyme ACC deaminase that cleaves ACC into ammonia and $\alpha$-ketobutyrate, thereby decreasing ACC in plants and ethylene in the leaves [67,69–71]. This further supports their use to protect against waterlogging stress.

### 4.3. Effect of Fungi on Plant Waterlogging Response

*Saccharomyces cervisiae* is a plant growth-promoting yeast (PGPY) that is used for soil fertilization or as a foliar application on plants [72,73]. Yeasts (such as *S. cervisiae*) act as natural stimulators and are rich in protein, carbohydrates, nucleic acid, lipids, minerals (Na, Fe, Mg, K, P, S, Zn, Mn, Cu, Si, Cr, Ni, Va, Li), thiamine, riboflavin, pyridoxine, hormones, biotin, B12, and folic acid [74]. In general, PGPY benefit host plants by influencing phytohormone production, improving soil fertility and nutrient availability, and enhancing abiotic stress resistance [75]. Other species of yeast, such as *Hansenula saturnus* and *Issatchenkia occidentalis*, have been found to produce ACC deaminase, which, as mentioned in the previous section, facilitates a plant's antioxidant system under waterlogged conditions [76,77]. Yeasts are also a direct source of enzymes and amino acids and are generally recognized as safe (GRAS) for field application [75], making them a useful tool for sustainable agricultural production.

Various studies support yeast's positive effect on plant growth, total yield, quality, chemical constituents, the formation of photosynthetic pigments, and increased antioxidant activity [72,78–80]. Similarly to bacteria, they also contain and promote the regulation of plant growth hormones such as auxins, gibberellins, and cytokinins [72,73,75,80,81]. As a general note, the cultivation of yeast with other PGPB enhances its ability to promote cell activation, root division, and vegetative growth [73,75]. This suggests that biostimulants containing fungi and bacteria together might be more effective than biostimulants containing only one of the organisms. Furthermore, it should be noted that positive effects of foliar application on plant growth are sometimes only seen at certain stages of development; for example, enhanced shoot growth was seen in the first stage of wheat growth but not later [72]. This should be taken into consideration when planning biostimulant application schedules.

Other types of fungi, such as *Penicillium citrinum* and *Trichoderma asperellum*, are also known to produce ACC deaminase [82,83], thereby assisting with ACC detoxification and minimizing damage to plants experiencing abiotic stress. *P. citrinum* is an endophytic

fungus and can be found, among other plants, in the rhizosphere of *Brassica rapa* var. *parachinensis* [84]. *P. citrinum* has growth promotion effects, assists nutrient assimilation, increases plant biomass, and releases gibberellins and cytokinins [84]. Two mechanisms for *P. citrinum* and other endophytes to counteract abiotic stress have been suggested: (1) By activating the host response system, and (2) by generating antistress metabolites [85]. Interestingly, microbes exposed to abiotic stress produce auxin, GA, and compatible solutes, which confer stress tolerance to plant hosts [86]. Since levels of phytohormones can become reduced in plants under stress, hormones produced by endophytes can benefit the plant through their antimicrobial properties and by acting as a source of nutrients [87]. Additionally, microbes produce antioxidants and accumulate ABA, which reduces the level of ROS in plants and allows for growth under conditions of stress [85].

Overall, fungi appear to benefit plants experiencing abiotic stress in similar ways to bacteria, through positive effects on nutrient status and phytohormone modulation. Therefore, biostimulants containing fungi could be found to benefit plants experiencing stress from waterlogged conditions.

### 4.4. Effect of Amino Acids on Plant Nutrients and Abiotic Stress

Amino acids are compounds that contain an amine functional group and a carboxylic acid functional group [88]. They serve various functions in plants, including biotic and abiotic stress protection, signalling, N storage, and the chelation of metals [89]. Commercially available amino acids tend to be short peptides known as protein hydrolysates, which have been isolated from plant, animal, or microorganism material [12]. Amino acid biostimulants can be applied foliarly and are absorbed by plants through the leaves via diffusion [90], or they can be absorbed by plant roots through specific transporters [91,92]. Amino acid soil inoculations have been found to have a positive effect on soil health by reinforcing microbial networking and suppressing disease-related fungi, which benefits crop productivity [93]. This positive interaction between amino acids and fungi suggests that biostimulants containing both of these might show enhanced beneficial effects.

The application of amino acids as biostimulants has demonstrated positive effects on plant growth, yield, and stress mitigation [88]. Amino acids, such as cysteine, glycine, and glutamic acid, are primary metabolites that can act as compatible solutes in cells experiencing abiotic stress [94]. The accumulation of compatible solutes is one of the basic strategies employed by plants for their protection and survival during stressful conditions [95]. Haghighi et al. [96] applied an amino acid mix containing L-cysteine, L-glycine, and L-glutamic acid (among others) to drought-stressed cabbage, with the effect of increased antioxidants and increased phenolic, protein, and proline concentrations. This type of effect may be beneficial under extended waterlogged conditions to combat oxidative stress. In another study, the foliar application of amino acid extract biostimulants enhanced soybean resistance to hypoxic stress, resulting in improved physiological parameters, plant development, and yield [97].

Some studies have found that amino acid application can improve nutrient uptake [98]. Exogenous amino acid application to plant leaves has been shown to increase uptake and nutrient-use efficiency of macro- and micronutrients, in particular N, Fe, Zn, Cu, and Mn [88]. Different crops vary in the way they respond to amino acid treatment; in some crops (apples and tomatoes), Ca uptake was improved with amino acid application; however, there was no effect in other crops (kiwifruit) [99,100]. The mechanism for amino acid improvement of plant nutrition via foliar application involves direct changes to the plant physiology: Improving nutrient mobility, changing root morphology, and increasing the activity of $NO_3$-assimilation enzymes [88]. Amino acids can also chelate metals (Fe, Zn, Mn, and Cu) and facilitate their absorption through the leaves (via specific amino acid transporters) and translocation within the plant [88,101,102]. These positive effects on nutrients could benefit plants under waterlogging stress, when nutrient uptake through the root system is inhibited.

## 5. Mineral Biostimulants (Nano-CaCO$_3$)

Some biostimulants, e.g., Eco Green, are based on mineral nanoparticles such as nano-CaCO$_3$. Nanotechnology, a form of precision agriculture, is becoming progressively more prevalent in crop production due to its ability to enhance crop productivity [103]. Plant products based on nanomaterials have several advantages over conventional products, including increased efficacy, reduced input requirements, and lower environmental toxicity [104]. When applied as an ameliorative agent against plant stress, the success of nano-products is due to their small size, which improves their contact with leaf pores and facilitates translocation in plant vessels [105]. Nano CaCO$_3$ has been applied as a foliar spray, dissolved in hydroponic solution, or applied as fertigation [106–108]. Foliar applications are beneficial to plants as they work rapidly and independently of soil conditions [106]. This is important if there is a need to apply biostimulants during waterlogged conditions, when soil is saturated and soil applications become diluted. In hydroponic conditions, nano-CaCO$_3$ application to the roots of wheat seedlings showed an increase in SOD and APX activities (54% and 58%, respectively) and significantly enhanced photosynthesis-related physiological indicators [107], demonstrating that this type of application may also be beneficial if applied once waterlogged conditions have receded.

Calcium (Ca$^{2+}$) is an important macronutrient for plant growth and development; it increases yield, nitrate absorption, chlorophyll content, transpiration and photosynthetic rates, and stomatal conductance [109,110]. Nano-CaCO$_3$ has been applied foliarly to various ornamental plants and agricultural crops, resulting in increases in plant size, dry and fresh weight, yield, and improved nutrient absorption in shoots [109,110]. Nano-CaCO$_3$ is said to improve plant productivity and increase plant immunity and resistance to pests, diseases, and environmental stress [111]. Furthermore, CaCO$_3$ penetrates into the leaf and is dissolved into calcium oxide (CaO) and then carbon dioxide (CO$_2$), enhancing photosynthesis in the leaf [111]. When applied to the roots of wheat, nano-CaCO$_3$ increased antioxidant activity, with more than 50% higher SOD and APX activities in treated plants compared to non-treated plants [107]. This indicates that the application of nano-CaCO$_3$ might be beneficial under waterlogged conditions to alleviate oxidative stress.

Currently, the body of research investigating nano-CaCO$_3$ as a nano-fertilizer is greater than that investigating it as a biostimulant. In particular, the nano-fertilizer Lithovit has been extensively studied on various crops. Lithovit is a natural CO$_2$ nano-fertilizer made from micronized calcite that acts as a source of Ca and Mg [112]. Due to their similar composition, the results from the foliar application of Lithovit might be relevant when investigating the effects of foliar nano-CaCO$_3$ biostimulant applications. When applied foliarly to cucumber, broccoli, *Echinacea purpurea*, soybean, and potato, the beneficial effects of Lithovit were numerous: increased vegetative growth characteristics (plant height, number of leaves, leaf area, and dry matter), increased chemical constituents (chlorophyll, carotenoids, total carbohydrates, nutrients), increased productivity and total yield, and increased total sugars and phenol compounds [106,112–115].

Regarding nutrients, Lithovit has been found to increase absorption and content of certain elements; the nutrients influenced depend on the crop involved. In chilli peppers, Ca and Mg content was significantly higher in the foliage of treated plants [105]. This is to be expected, due to Lithovit containing Ca and Mg and acting as a foliar fertilizer. On salt-stressed tomato plants, Lithovit contributed to better absorption of Ca, Mg, Si, Fe, and Mn [108]. Once again, this is believed to be due to Lithovit containing Ca, Mg, and micronutrients such as Fe and Mn. In cucumber, Lithovit increased N, P, K, Ca, Zn, and B in the foliage [113]. In *Echinacea purpurea*, Lithovit increased N, P, and K content [115], while in soybean foliage, the content of N, P, K, Ca, Mg, and Fe increased [114]. The results of these studies demonstrate how the effect on nutrients of nano-CaCO$_3$ application varies for different crops; further studies would be needed to investigate the effect of nano-CaCO$_3$ biostimulant application on plant nutrients in cabbage. The enhancement of antioxidant activity and photosynthesis gives evidence to support the use of nano-CaCO$_3$

in plants experiencing oxidative stress, suggesting that further experimental studies would be worthwhile.

## 6. Conclusions

A large variety of natural substances can act as biostimulants to improve plant health, growth, and productivity. These substances are increasingly being investigated and formulated for their use in agricultural production and are compatible with the transition to a more sustainable agricultural production system. With a focus on *Brassica oleracea* var. *capitata*, this review investigates the potential of applying biostimulants based on algae extracts, amino acids, beneficial microorganisms, and nano-$CaCO_3$ to waterlogged plants to reduce the negative effects associated with anoxic stress. The studies reviewed have shown that biostimulants based on these products can enhance nutrient availability and plant nutrient status, modulate phytohormones and phytohormone signalling, increase the concentration of compatible solutes, and improve antioxidant activity. These effects result in improved plant tolerance to abiotic stress and generally better plant productivity and growth, indicating that their use to combat waterlogging stress could be beneficial and worthwhile. However, the studies have clearly shown that multiple variables impact the way a plant responds to biostimulant application, namely plant species and development stage, growing conditions, and type of application (foliar, soil inoculation). Furthermore, although some experiments have taken place under field conditions, many experiments have taken place under strictly controlled laboratory conditions, and it is not known whether the same results would be seen outdoors. Similarly, in many experiments, plants were not grown to maturity and harvested, so there is less information available as to how the beneficial effects of application at an early stage would impact the final yield, quality, and other economically relevant characteristics of crops.

The positive results of biostimulant applications to stressed plants are significant enough to warrant further investigation, especially since their status as natural products gives them an advantage over conventional products as the agriculture sector transitions to more sustainable and ecological management. Since biostimulants can be applied foliarly or as a soil inoculation, this gives flexibility to their use, and it is possible to apply them during flood conditions and/or afterwards. Further field studies are required to evaluate and quantify the benefits of biostimulant application in a real-life setting to alleviate anoxic stress in waterlogged plants and to determine the most appropriate type, timing, and rate of application. In addition, future studies might also like to investigate the most effective biostimulant combinations, since beneficial interactions have been noted between different microorganisms, fungi, and amino acids. This type of research could provide valuable practical guidance to food producers in Europe and globally, as well as give them additional tools to assist with adaptation to changing climate conditions.

**Author Contributions:** Conceptualization, M.P. and N.B.; methodology, M.P.; formal analysis, N.B.; investigation, N.B.; data curation, N.B.; writing—original draft preparation, N.B.; writing—review and editing, M.P.; visualization, N.B.; supervision, M.P.; funding acquisition, M.P. All authors have read and agreed to the published version of the manuscript.

**Funding:** Publication was supported by the OpenAccess Publication Fund of the University of Zagreb Faculty of Agriculture.

**Institutional Review Board Statement:** Not applicable.

**Data Availability Statement:** No new data were created or analysed in this study. Data sharing is not applicable to this article.

**Conflicts of Interest:** The authors declare no conflict of interest.

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
