# Peer review of "Use of Biostimulants to Alleviate Anoxic Stress in Waterlogged Cabbage (Brassica oleracea var. capitata)—A Review"

_agriculture, doi:10.3390/agriculture13122223_

Round 1

Reviewer 1 Report

Comments and Suggestions for Authors

I have had the pleasure of reviewing your manuscript titled " Use of biostimulants to alleviate anoxic stress in waterlogged cabbage (Brassica oleracea var. capitata) – a review." I must commend your comprehensive and robust approach to this pertinent issue in the realm of agricultural science.

The issue of biostimulants application for abiotic alleviation is indeed a pressing one, and your review not only highlights the problem but provides a tangible and innovative solution. However, there are some crucial amendments required as follow:

Point 1: The title of the review, "Use of biostimulants to alleviate anoxic stress in waterlogged cabbage (Brassica oleracea var. capitata) – a review," warrants some constructive criticism as it may not accurately represent the comprehensive scope of the review. The article, in fact, delves into the applications of biostimulants for alleviating anoxic stress in a variety of crops, not just limited to cabbage. Therefore, the title's specificity may inadvertently deter potential readers interested in biostimulant applications for other crops. To better reflect the article's broader content and to make it more inclusive, a revised title such as "Biostimulant Applications for Alleviating Anoxic Stress in Waterlogged Crops: A Comprehensive Review" would be more suitable, maintaining clarity while encompassing the wide range of crop types discussed within the review. This revision would help attract a broader readership and ensure that the title accurately mirrors the article's content.

Point 2: My second comment pertains to the Abstract section. The abstract in its current form appears to be concise to the point of brevity, lacking a comprehensive overview of the article's content. A more comprehensive abstract is crucial to provide potential readers with a clear understanding of the review's objectives, methodology, key findings, and implications. Expanding the abstract to include a brief introduction to the context of increasing flooding events in Europe due to climate change and the resulting challenges in agriculture would set the stage for the reader. Furthermore, including a summary of the methodologies employed in reviewing the studies and a concise overview of the significant biostimulants discussed, such as algae extracts, amino acids, microorganisms, and nano CaCO3, and their respective effects on plant responses to waterlogged conditions, would offer a more informative glimpse into the article's content. Additionally, a statement regarding the practical implications or potential applications of these findings for agriculture would help readers grasp the article's significance. Thus, expanding the abstract with these elements would enhance its comprehensiveness and provide a better representation of the article's scope and relevance.

Point 3: Moreover, it is essential to incorporate a section in the discussion that highlights the recent utilization of natural extracts as environmentally friendly biostimulants, such as Moringa leaf extract, biostimulants derived from natural bee honey, and extracts from fennel and ammi seeds. These emerging biostimulants have gained recognition for their potential benefits in agriculture, and their inclusion in the discussion would enhance the article's relevance by acknowledging the evolving landscape of biostimulants. The following publications may prove to be valuable resources for your research.

[1] E.M. Desoky, L.M.M. El-maghraby, A.E. Awad, A.I. Abdo, M.M. Rady, W.M. Semida, Fennel and ammi seed extracts modulate antioxidant defence system and alleviate salinity stress in cowpea ( Vigna unguiculata ), Sci. Hortic. (Amsterdam)., 272 (2020) 109576.

[2] H.F. Alharby, H.S. Al-Zahrani, K.R. Hakeem, H. Alsamadany, E.S.M. Desoky, M.M. Rady, Silymarin-enriched biostimulant foliar application minimizes the toxicity of cadmium in maize by suppressing oxidative stress and elevating antioxidant gene expression, Biomolecules, 11 (2021) 1–28.

Reviewer 2 Report

Comments and Suggestions for Authors

Use of biostimulants to alleviate anoxic stress in waterlogged cabbage (Brassica oleracea var. capitata) – a review by Nadya Buga*, Marko Petek.

The title of this manuscript is interesting to read and in the scope of Agriculture.

The review in each section is look good, but the content shows less information of anoxic stress in Brassica oleracea, and did not present the way or methods and doses of biostimulants for the users.

I would like to suggest the authors to focus more on “Brassica, anoxia, and biostimulant” how are they related in this review, and provide in-depth discuss.

Some comments and suggestions below:

Introduction,

1. As the authors mentioned the importance of cabbage in Europe. The paper needs more information of “cabbage” in Europe, such as the areas, varieties, consumption quantity…

And more information about Brassica oleracea var. capitata. How to cultivate it: types of soil, water volume?

2. The effects of flood or waterlogging on cabbage, or vegetables in Europe are needed (only line 29-34 seen).

3. The data of heavy rain in Europe needs to explain how it will affect the cabbage cultivation. How much of flood level will be?

4. Please provide the threshold of flood for cabbage (line 38-37, 103 – intolerant species)

 Discussion

2.1 Flooding and Nutrients (line 68)

- please explain about the nutrients related to flooding

2.2 line 97-133

- please provide the research and results on the effects of hypoxia or anoxia on Brassica oleracea.

- the effects of hypoxia or anoxia on Brassica oleracea

2.3 line 147-154

- Are the applications of ANEs on Brassica under hypoxia or anoxia?

- The authors need to explain more detail of [25], doses used, application methods, ages of Brassica, the situation of hypoxia/anoxia, duration after application before compound analysis.

2.3.1 line 180-182

- - The authors need to explain more detail of [31], doses used, application methods, ages of Brassica, the situation of hypoxia/anoxia, duration after application before compound analysis.

Conclusion

-        Need more focus for the reader.

Reviewer 3 Report

Comments and Suggestions for Authors

I have evaluated the manuscript entitled “Use of biostimulants to alleviate anoxic stress in waterlogged cabbage (Brassica oleracea var. capitata)” The idea of the study is interesting; however, there are some comments. Kindly find the attached file.

Comments on the Quality of English Language

Minor editing of English language required.

Round 2

Reviewer 2 Report

Comments and Suggestions for Authors

The author responses are well prepared, and the revised manuscript is scientifically improved. 

Reviewer 3 Report

Comments and Suggestions for Authors

Well done.